



1       February 13, 2020

## 4    Brief communication:

## 5    Mapping Greenland's perennial firn aquifers using enhanced-

## 6    resolution L-band brightness temperature image time series

*Julie Z. Miller[1,2], David G. Long[3], Kenneth C. Jezek[4], Joel T., Johnson[5],*
*Mary J. Brodzik[1,6], Christopher A. Shuman[7], Lora S. Koenig[1,6], & Theodore A. Scambos[1,2]*
[1]*Cooperative Institute for Research in Environmental Sciences, University of Colorado, Boulder, Colorado, USA*
[2]*Earth Science and Observation Center, University of Colorado, Boulder, Colorado, USA*
[3]*Department of Electrical and Computer Engineering, Brigham Young University, Provo, Utah, USA*
[4]*Byrd Polar and Climate Research Center, The Ohio State University, Columbus, Ohio, USA*
[5]*Department of Electrical and Computer Engineering, The Ohio State University, Columbus, Ohio, USA*
[6]*National Snow and Ice Data Center, University of Colorado, Boulder, Colorado, USA*
[7]*University of Maryland, Baltimore County, NASA Godard Space Flight Center, Greenbelt, Maryland, USA*

**Correspondence to: jzmiller.research@gmail.com**



## 23    Abstract

Enhanced-resolution L-band brightness temperature ($T_B$) image time series collected over the
Greenland ice sheet by NASA's Soil Moisture Active Passive (SMAP) satellite are used to map
Greenland's perennial firn aquifers from space. Exponentially decreasing L-band $T_B$ signatures
are correlated with perennial firn aquifer areas identified via the Center for Remote Sensing of Ice
Sheets (CReSIS) Multi-Channel Coherent Radar Depth Sounder (MCoRDS) flown by NASA's
Operation IceBridge (OIB) campaign. An empirical algorithm to map extent is developed by fitting
these signatures to a set of sigmoidal curves. During the spring of 2016, perennial firn aquifer
areas are found to extend over ~66,000 km$^2$.

## 33    1    Introduction

Firn is a porous layer of recrystallized snow near the surface of a glacier or an ice sheet. Under
certain climate conditions, firn can host a laterally unconfined aquifer and thereby buffer meltwater
flow across the near-surface to the periphery of the ice. Given sufficiently high surface melting,





firn aquifers form, or recharge, in the percolation facies as a result of vertical and lateral
percolation of meltwater into the pore space of surface snow, if present, and firn layers overlying
impermeable ice layers. If a firn aquifer is adjacent to, or is crossed by, a crevasse, mobile
meltwater can initiate meltwater-driven hydrofracture, drain into the subglacial hydrological
system, and accelerate ice flow (Fountain and Walder, 1998).

Although common on glaciers, firn aquifers were unknown on ice sheets until their discovery
during an April 2011 field expedition to the percolation facies of southeastern Greenland (Forster
et al., 2014). Greenland's firn aquifers store meltwater seasonally, intermittently, or perennially,
depending on location and climate. They range from shallow water-saturated firn layers that perch
on top of near-surface ice layers, to deeper water-saturated firn layers that extend from the ice
sheet surface to the firn-ice transition. Greenland's perennial firn aquifers are expansive (their
simulated extent ranges between 55,700 km$^2$ and 90,200 km$^2$ (2010-2014); Steger et al., 2017),
and are capable of storing substantial volumes of meltwater (~140±20 GT; Koenig et al., 2014).
Simulations using a simple firn model suggest that high snowfall thermally insulates water-
saturated firn layers, allowing meltwater to be stored in liquid form throughout the freezing season
(i.e., the time between surface freeze-up to melt onset) if the overlying snow layer is sufficiently
deep (Munneke et al., 2014). Perennial firn aquifers have also recently been discovered off the
coast of western Greenland on the Maniitsoq (Forster et al., 2014) ice cap, and on a Svalbard
icefield (Christianson et al., 2015).

The existence and approximate extent of Greenland's perennial firn aquifers has been
demonstrated using shallow firn cores extracted from several sites in southeastern Greenland
during recent field expeditions, (Forster et al., 2014; Koenig et al., 2014; Miller et al., 2017a), and
ice-penetrating radar surveys collected by the CreSIS Accumulation Radar flown by NASA's
2010-2014 OIB campaign (Miège et al., 2016). However, these airborne observations are not
comprehensive over the Greenland ice sheet, occur only during the spring months, and are often
non-repeating in space and time. The 2019 conclusion of NASA's OIB campaign ended even
these sparse mapping activities. Thus, this leaves significant gaps where Greenland's perennial
firn aquifers are yet to be mapped (Fig. 1).

Here, we demonstrate the potential for mapping Greenland's perennial firn aquifers from space
using satellite L-band microwave radiometry. We use recently released enhanced-resolution L-
band $T_B$ image time series collected over the Greenland Ice Sheet by the microwave radiometer





aboard the SMAP satellite (Long et al., 2019) together with coincident thermal infrared (TIR) $T_B$
image time series collected by the Moderate Resolution Imaging Spectroradiometer (MODIS)
aboard the Terra and Aqua satellites (Hall et al., 2012) to develop an empirical algorithm to map
extent.

## 76  2      Methods

### 77  2.1    Enhanced-resolution L-band $T_B$ image time series

The SMAP satellite was launched 1 January 2015 and carries a microwave radiometer that
operates at a frequency of 1.41 GHz (L-band). It is currently collecting global observations of
vertically and horizontally (V- and H-pol, respectively) polarized $T_B$. The surface incidence angle
is ~40°, the radiometric accuracy is ~1.3 K, and the resolution is ~40 km.

The microwave radiometer form of the Scatterometer Image Reconstruction (rSIR) algorithm
generates $T_B$ on a fine spatial grid using satellite observations (Long et al., 2019). The rSIR
algorithm exploits the spatial measurement response function (MRF) for each observation, which
is a smeared version of the antenna pattern. Using the overlapping MRFs, the rSIR algorithm
reconstructs $T_B$ from the spatially filtered, low-resolution sampling provided by the satellite
observations. In effect, it generates an MRF deconvolved $T_B$ image. Combining multiple passes
increases the sampling density, which further improves the accuracy and resolution of the rSIR
reconstruction.

Given converging orbital passes in the polar regions, the SMAP satellite passes over Greenland
several times each day, and provides nearly complete coverage during two distinct local time-of-
day intervals. The rSIR algorithm combines orbital passes that occur between 8 a.m. and 4 p.m.
(+/-2 hours) local time-of-day to reconstruct twice-daily (morning and evening orbital pass interval,
respectively) $T_B$ images. $T_B$ image data are projected on the Equal-Area Scalable Earth Grid
(*EASE-Grid 2.0*) (Brodzik et al., 2012) at a 3.125 km grid cell spacing. The effective resolution for
each grid cell is dependent on the number of satellite observations used in the reconstruction.

Fig. 1 shows an MRF deconvolved (3.125 km) $T_B$ image projected on the Northern Hemisphere
*EASE-Grid 2.0* over Greenland. The image was reconstructed using H-pol $T_B$ observations
collected during the evening orbital pass interval on 15 April 2016, coincident with airborne ice-
penetrating radar surveys. The enhanced-resolution image clearly captures many ice sheet

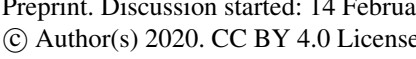


features, particularly in the percolation facies where perennial firn aquifer areas have been
mapped (Fig. 1a, b).

Fig. 1 also includes a Phased Array type L-band Synthetic Aperture Radar (ALOS-PALSAR)
winter season radar backscatter ($\sigma^o$) image mosaic (Joughin et al., 2016). ALOS-PALSAR
operates at a frequency of 1.27 GHz. Image data are collected in H-pol transmit and receive mode
(HH), with surface incidence angle between 36.4° and 40.8°. As observed by active satellite
microwave instruments (i.e., synthetic aperture radar and scatterometry), the percolation facies
of ice sheets have long been known to exhibit some of the highest $\sigma^o$ magnitudes on Earth (Jezek
et al., 1993; Long and Drinkwater, 1994). Refreezing of seasonal meltwater results in the
formation of an intricate network of embedded ice structures (i.e., ice pipes, lenses, and layers)
that are large relative to these instruments centimeter wavelengths (~10-100 cm long, ~10-20 cm
wide) and induce strong volume scattering. The bright white regions of very high (uncalibrated)
$\sigma^o$ represent Greenland's percolation facies (Fig. 1c, d). During the freezing season, $T_B$ over the
percolation facies is inversely related to $\sigma^o$. $T_B$ magnitudes on the Greenland ice sheet are lowest
over the percolation facies, with values ranging from ~130 K to 230 K (V-pol channel) and ~100
K to 200 K (H-pol channel).

We analyzed SMAP 2015-2016 morning and evening V- and H-pol $T_B$ images together with
coincident MODIS 2016 TIR $T_B$-derived surface meltwater maps (described in section 2.2) in the
percolation facies. Although sharp increases in $T_B$ delineate the extent of surface melting and
provide a qualitative indication of the volumetric fraction of near-surface meltwater, a distinct
subsurface meltwater signal is not easily distinguishable in any of the $T_B$ images during the
freezing season.

To discriminate between L-band perennial firn aquifer emissions and background ice sheet
emissions, we used image time series analysis. We used the Greenland Ice Mapping Project
(GIMP) Land Ice and Ocean Classification Mask (Howat et al., 2014) to construct an ice-masked
V- and H-pol $T_B$ image time series that alternates morning and evening orbital pass observations
collected between 31 March 2015 (i.e., the beginning of the SMAP data record) and 31 December
2016. These image time series provide sufficient detail to analyze spatiotemporal differences in
exponentially decreasing L-band $T_B$ signatures over perennial firn aquifer areas as compared to
other percolation facies areas where seasonal meltwater is stored as embedded ice.

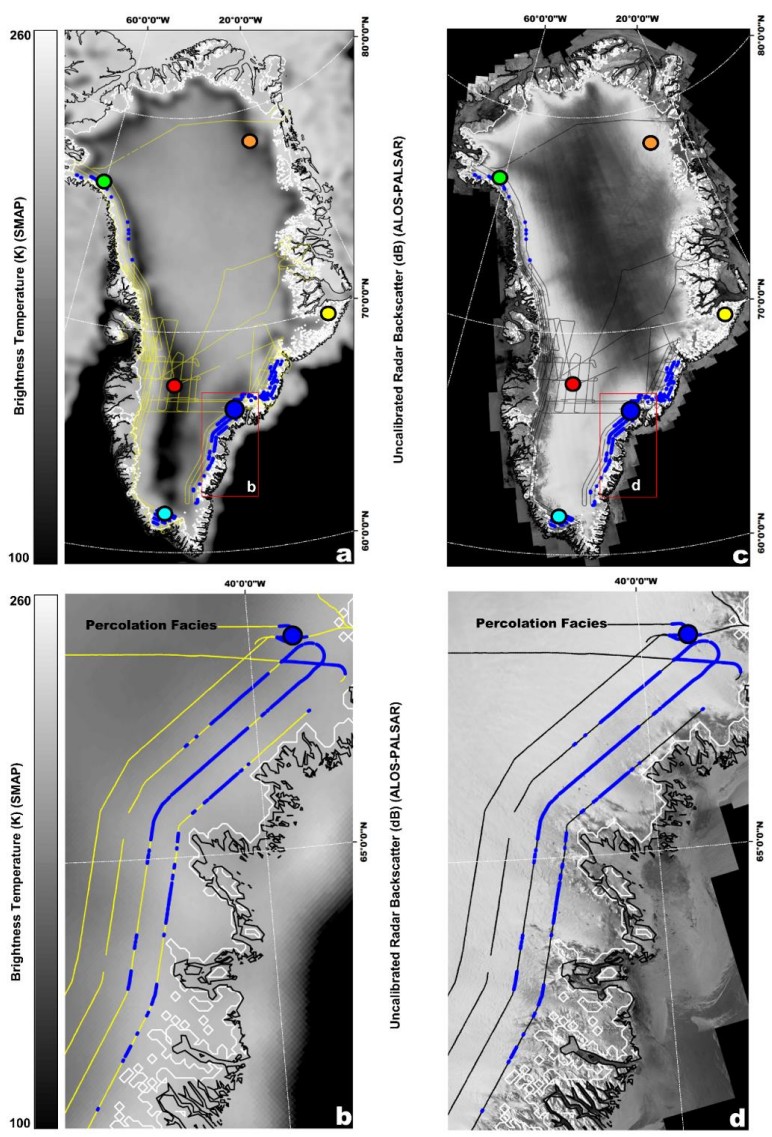

**Figure 1**
*(a) Enhanced-resolution (3.125 km) H-pol L-band $T_B$ image collected 15 April 2016 by the*
*microwave radiometer aboard the SMAP satellite during the evening orbital pass interval over*
*Greenland (Long et al., 2019); and (b) high-resolution (100 m) ALOS-PALSAR winter season*
*(April 2009-April 2010) HH σ° image mosaic (Joughin et al., 2016). Black regions of low L-band*
*$T_B$ magnitudes and bright white regions of very high (uncalibrated) L-band σ° magnitudes are the*
*percolation facies. Zoom areas (red boxes, b and d) are 2016 MCoRDS-derived perennial firn*
*aquifer locations (blue circles) in the southeastern percolation facies along OIB flight lines (yellow*
*and black lines, respectively). Test Site 1 (blue circle); Test Site 2 (cyan circle); Test Site 3 (red*
*circle); Test Site 4 (green circle); Test Site 5 (orange circle); Test Site 6 (yellow circle); GIMP-*
*derived ice extent (white lines, Howat et al., 2014); and coastlines (black lines, Wessel and Smith,*
*1996). Test Site 1 is a shallow firn core site in southeastern Greenland.*



**2.2    Thermal infrared T$_B$-derived surface freeze-up and melt onset dates**
The Ice Surface Temperature (IST) algorithm (Hall et al., 2012) retrieves a clear-sky ice surface
skin temperature (i.e., temperature at radiative equilibrium) over the Greenland ice sheet accurate
within ±1°C using a split window technique and satellite observations collected by MODIS TIR T$_B$
channels 31 (10.78 μm–11.28 μm) and 32 (11.77 μm–12.27 μm) during daily orbital passes that
occur between 12 a.m. and 12 p.m. local-time-of-day. The resolution is 0.78 km. For temperatures
that are close to 0°C, IST values are closely compatible with contemporaneous NOAA near-
surface air temperature data (Shuman et al., 2014). IST data use the standard MODIS 1 km
resolution cloud mask ('MOD35') that uses up to 14 spectral bands and multiple spectral and
thermal tests to identify clouds.

We projected the IST image data onto the *EASE-Grid 2.0* at a grid cell spacing of 3.125 km, and
then used the GIMP mask (Howat et al., 2014) to construct ice-masked IST image time series
between 31 March 2015 and 31 December 2016. Using the IST image time series, we retrieved
surface meltwater maps for the 2015 and 2016 melting season. We set a threshold of IST ≥ -1°C
for surface meltwater detection, consistent with the ±1°C accuracy of the IST image data (i.e.,
surface meltwater is inferred when IST is as low as -1°C). This threshold represents a theoretical
penetration depth from ~2 μm beneath the snow. We constructed a 2015 surface meltwater mask
by marking each grid cell in which surface meltwater was detected in at least one time step. We
estimated the 2015 surface freeze-up date for each grid cell as the time step following the last
time step at which surface meltwater was detected. For each grid cell that melted in 2015, we
estimated the 2016 melt onset date as the first time step at which surface meltwater was detected.

**2.3    Airborne ice-penetrating radar surveys**
In April and May of 2016, the MCoRDS instrument was flown over the Greenland ice sheet and
its peripheral ice caps aboard a NOAA P-3 aircraft as part of NASA's OIB campaign. Perennial
firn aquifer locations were detected using radar echograms collected by the MCoRDS instrument
and the methodology described in Miège et al., (2016). Strong, non-conformal, upper reflectors in
radar echograms are interpreted as the upper surface of stored meltwater. The total number of
mapped perennial firn aquifer locations is 78,343 (Fig. 1). We projected these perennial firn
aquifer locations on the *EASE-Grid 2.0* at a grid cell spacing of 3.125 km. The total number of
grid cells with at least one perennial firn aquifer location is 780 corresponding to 7922 km$^2$.



### 2.4 Algorithm

### 2.4.1 L-band perennial firn aquifer signatures

We analyzed V- and H-pol $T_B$ time series over perennial firn aquifer areas identified via airborne ice-penetrating radar surveys. These time series were overlaid with TIR $T_B$-derived surface freeze-up and melt onset dates to partition the freezing season. Throughout the percolation facies, L-band $T_B$ signatures over perennial firn aquifer areas exhibit relatively slow (i.e., time scales of ~months) exponential decreases that approach or achieve relatively stable magnitudes late in the freezing season. In contrast, L-band $T_B$ signatures over other percolation facies areas where seasonal meltwater is stored as embedded ice, exhibit relatively rapid (time scales of ~weeks to days) exponential decreases, subsequently achieve relatively stable magnitudes early in the freezing season, and remain relatively stable until melt onset the following year. Spatiotemporal differences in exponentially decreasing L-band $T_B$ signatures are used to detect additional perennial firn aquifers locations.

### 2.4.2 Continuous logistic model

We seek a simple mathematical relation that can be fit to the exponentially decreasing L-band $T_B$ signatures observed over the percolation facies. The continuous logistic model satisfies this requirement. It is based on a differential equation that models the increase or decrease in many types of physical systems as a set of simple S shaped or 'sigmoidal' curves. These curves begin with an initial interval of increase or decrease that is approximately exponential. Then, as the function approaches its limit, the increase or decrease slows to approximately linear. Finally, as the function asymptotically reaches its limit, the increase or decrease exponentially tails off and achieves stable values. The continuous logistic model is described by a differential equation known as the logistic equation

$$\frac{dx}{dt} = \zeta x(1-x), \tag{1}$$

that has the solution

$$x(t) = \frac{1}{1+\left(\frac{1}{x_o}-1\right)e^{-\zeta t}}, \tag{2}$$

where $x_o$ is the function's initial value, $\zeta$ is the function's exponential rate of brightness temperature increase or decrease, and $t$ is time. The function $x(t)$ is also known as the sigmoid function. We use the sigmoid function to model the observed exponentially decreasing $T_B$ signatures as a set of decreasing sigmoidal curves. To simplify the analysis, the $T_B$ time series for each grid cell is first normalized





$$T_{B,N} = \frac{T_B - T_{min}}{T_{max} - T_{min}}. \qquad (3)$$

and the sigmoid fit is then applied as
$$T_{B,N}\big(t \in [t_{sfu}, t_{mo}]\big) = \frac{1}{1 + \left(\frac{1}{T_{B,N}(t_{sfu})} - 1\right)e^{-\zeta t}}. \qquad (4)$$

Here $T_{B,N}\big(t \in [t_{sfu}, t_{mo}]\big)$ is the normalized brightness temperature during the freezing season on
the time interval $t \in [t_{sfu}, t_{mo}]$, where $t_{sfu}$ is the surface freeze-up date, and $t_{mo}$ is the melt onset
date. The initial normalized brightness temperature is the function's value at the surface freeze-
up date, $T_{B,N}(t_{sfu})$, while the final normalized brightness temperature is the function's value at the
melt onset date $T_{B,N}(t_{mo})$. Note that $T_{B,N}(t_{mo})$ can be set by an appropriate selection of the
exponential rate of normalized brightness temperature increase or decrease. This parameter is
also used to distinguish between perennial firn aquifer areas and percolation facies areas where
seasonal meltwater is stored as embedded ice.

An example set of simulated sigmoidal curves generated using eq. 3 and eq. 4 is shown in Fig.
2a. For these simulated curves, the normalized brightness temperature at surface freeze-up is
fixed at a value of $T_{B,N}(t_{sfu}) = 0.9$, and the freezing-season duration is set to a value of $t \in [t_{sfu},$
$t_{mo}] = 300$ days, which is within the TIR $T_B$-derived freezing season duration range $t = [178$ days,
364 days]. The function's exponential rate of normalized brightness temperature decrease is set
to values between $\zeta = [-1, 0]$, incremented by time steps of 0.004. This interval represents our
model of exponentially decreasing L-band $T_B$ signatures over Greenland's percolation facies. The
blue lines correspond to the interval $\zeta \in [-0.04, -0.008]$, and produce curves similar to those
over perennial firn aquifer areas identified via airborne ice-penetrating radar surveys. This interval
is used to calibrate the algorithm. The grey lines correspond to the interval $\zeta \in [-1, -0.04)$, and
produce curves similar to those over percolation facies areas where seasonal meltwater is stored
as embedded ice.

**2.5    L-band $T_B$-derived perennial firn aquifer maps**
The curve fitting algorithm proceeds by smoothing V-pol $T_B$ time series that have been partitioned
by TIR $T_B$-derived surface freeze-up and melt onset dates, and then iteratively applying the
sigmoid fit. The V-pol channel exhibits decreased sensitivity to changes in the volumetric fraction
of meltwater attributed to reflection coefficient differences between the instrument's channels.
This results in a reduced chi-squared error statistic when fitting V-pol $T_B$ time series to the sigmoid
function. Surface freeze-up is fixed at a value of $T_{B,N}(t_{sfu}) = 0.9$. Fixing this parameter provides





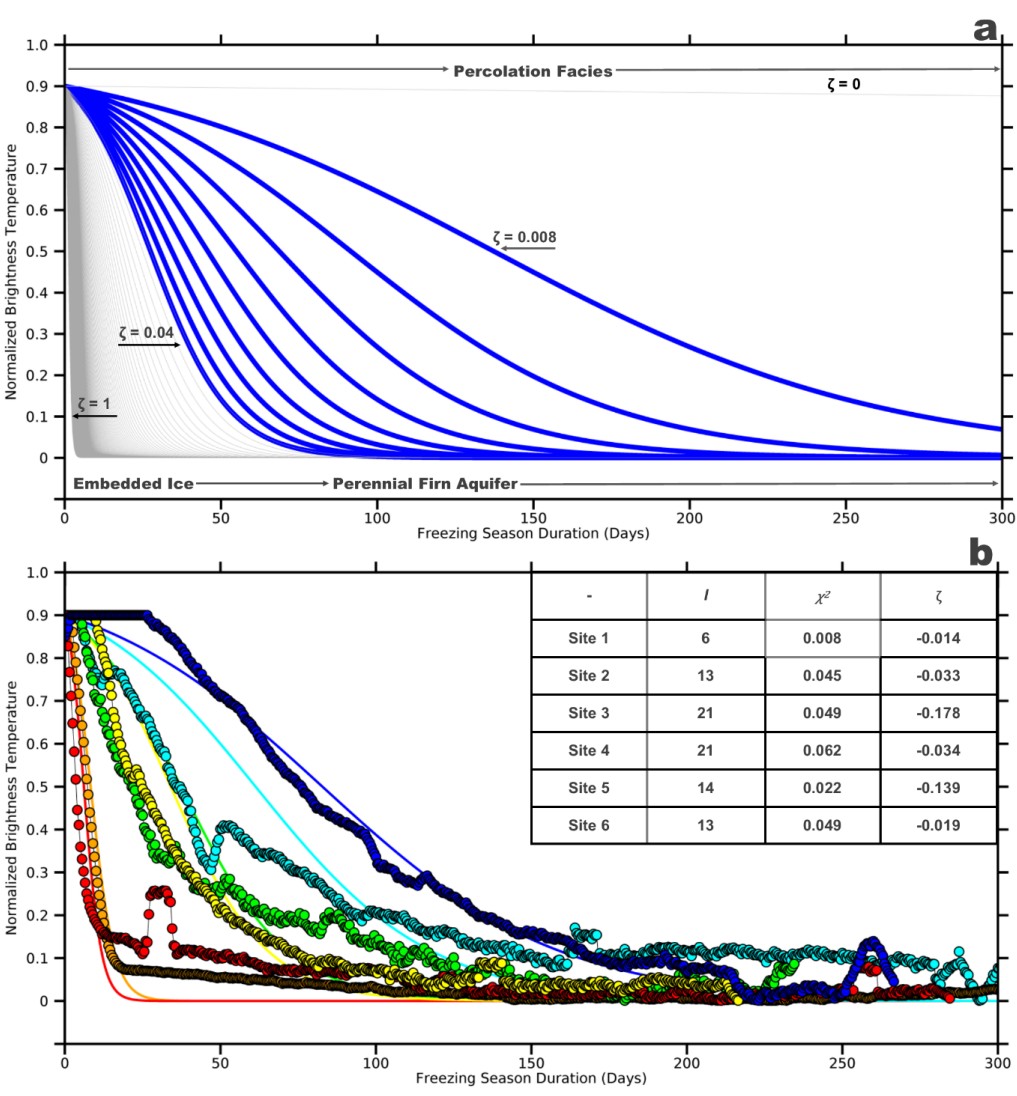


**Figure 2**
*(a) Example set of simulated sigmoidal curves corresponding to the interval ζ ∈ [−1, 0] which represents our model of exponentially decreasing L-band $T_B$ signatures over Greenland's percolation facies. Blue lines correspond to the calibration interval ζ ∈ [−0.04, −0.008] used to map Greenland's perennial firn aquifers; grey lines correspond to the interval ζ ∈ [−1, −0.04] which represents percolation facies areas where seasonal meltwater is stored as embedded ice. (b) Examples of $T_B$ time series over Test Sites 1-6 that have been iteratively fit to the sigmoid function using the curve fitting algorithm. Test Site 1 (blue circles and line); Test Site 2 (cyan circles and line); Test Site 3 (red circles and line); Test Site 4 (green circles and line); Test Site 5 (orange circles and line); Test Site 6 (yellow circles and line); associated curve fitting parameters (i.e., I, χ2, ζ)(upper right table).*






a uniform parameter space in which we could simply analyze the exponential rate of normalized
brightness temperature decrease. Although several fixed parameters were tested, this value
minimized the influence of meltwater detection (i.e., sharp increases in $T_B$ time series) resulting
from observational gaps and cloud contamination in the surface freeze-up dates, and provided
more  robust curve fitting. If the exponential rate of normalized brightness temperature decrease
is within the calibration interval, it is converted to a simple binary mapping parameter. $T_B$ time
series iteratively fit to the sigmoid function converge quickly (i.e., algorithm iterations $I \in [1, 10]$),
and satellite observations are a good fit (i.e., chi squared error statistic is  $\chi2 \in [0, 0.06]$),
indicating our algorithm provides a plausible satellite-derived map of the extent of Greenland's
perennial firn aquifers.

Fig. 2b illustrates examples of $T_B$ time series over Test Sites 1-6 (Fig. 1) that have been iteratively
fit to the sigmoid function using the curve fitting algorithm. Test Sites 1-4 (blue, teal, yellow, and
green, circles and lines) are examples of the relatively slowly exponentially decreasing L-band $T_B$
signatures exhibited over perennial firn aquifers areas. A shallow firn core site in southeastern
Greenland where meltwater was found stored at depths of 10 m and 25 m throughout the freezing
season (Forster et al., 2014) is within the Test Site 1 (blue circles and line) grid cell (Fig. 1b). Test
Sites 4 and 5 (orange and red circles and lines) are examples of the relatively rapidly exponentially
decreasing L-band $T_B$ signatures exhibited over other percolation facies areas. The associated
curve fitting parameters (i.e., $I$, $\chi2$, $\zeta$) for each of the Test Sites are given in the upper right table.

**3     Results**
Fig. 3a shows maps generated by the curve fitting algorithm over the Greenland ice sheet and its
peripheral ice caps. The GIMP-derived ice extent (~1.8 x $10^6$ km$^2$) is delineated via the peripheral
white line. During the 2015 melting season, the seasonal TIR $T_B$-derived surface meltwater extent
(~1.0 x $10^6$ km$^2$) is delineated via the red lines, and extends over ~57% of the total ice extent.
During the spring of 2016, the L-band $T_B$-derived perennial firn aquifer extent (66,000 km$^2$) is
mapped in blue, and extends over ~7% of the seasonal surface meltwater extent, and ~4% of the
total ice extent. Previously unknown perennial firn aquifer areas are mapped in northwestern
Greenland along the Lauge Koch Coast, in southern Greenland near Narsaq, in southeastern
Greenland along the King Frederick VI Coast and in King Christian IX's Land (Fig. 3b), and in
central east Greenland along the Blosseville Coast, on the Geike Plateau, and in King Christians
X's Land.

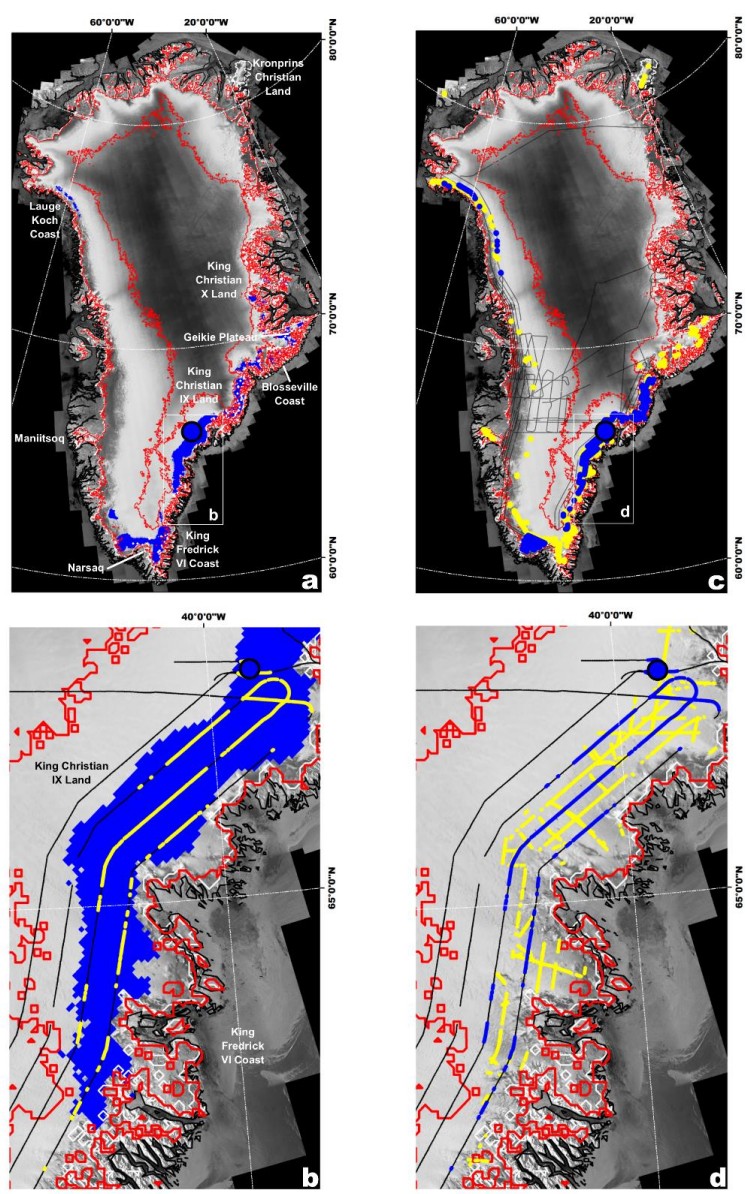

**Figure 3**

*(a) 2015-2016 SMAP L-band $T_B$-derived extent of Greenland's perennial firn aquifers (blue shading) overlaid on a high-resolution (100 m) ALOS-PALSAR winter season (April 2009-April 2010) HH σ° image mosaic (Joughin et al., 2016); (b) MCoRDS- (blue circles) and Accumulation Radar-derived (yellow circles, Miège et al., 2016) perennial firn aquifer locations along OIB flight lines (black lines); Zoom areas (white boxes, b and d) are 2016 MCoRDS- (yellow and blue circles, respectively) and Accumulation Radar-derived (yellow circles, Miège et al., 2016) perennial firn aquifer locations in the southeastern percolation facies; MODIS TIR $T_B$-derived surface meltwater extent (red line); and coastlines (black lines, Wessel and Smith, 1996).*



Fig. 3c shows airborne ice penetrating radar surveys. The blue circles are 2016 MCoRDS-derived
perennial firn aquifer areas. The black lines are NASA's 2016 OIB campaign flight lines. The
yellow circles are 2010-2014 Accumulation Radar-derived perennial firn aquifer locations (Miège
et al., 2016). The 2016 L-band $T_B$-derived perennial firn aquifer extent is consistent with the 2016
MCoRDS-derived locations. Exceptions include scattered locations near the ice extent edge (Fig.
3d) and along the upper perennial firn aquifer boundary (Fig. 3b). L-band ice sheet emissions are
likely mixed in these grid cells. Near the ice extent edge, perennial firn aquifer emissions are
influenced by morphological features, such crevasses and exposed glacial ice, and mix with
emissions from rock, land, and the ocean. Along the upper perennial firn aquifer boundary,
emissions are mixed with emissions from adjacent percolation facies areas where seasonal
meltwater is stored as embedded ice. The 2016 perennial firn aquifer extent is also consistent
with 2010-2014 Accumulation Radar-derived perennial firn aquifer locations, indicating that they
are multi-year ice sheet features in these areas. Exceptions include small, isolated locations in
northwestern, southwestern, and southern Greenland, the Maniitsoq Ice Cap, and scattered
locations near the ice extent edge and along the upper perennial firn aquifer boundary. Locations
near the ice extent edge possibly drained into crevasses, or were refrozen as superimposed ice.

**4      Summary and future work**
Our results indicate satellite L-band microwave radiometry is an effective tool for mapping the
extent of Greenland's perennial firn aquifers. We have derived an empirical algorithm by analyzing
spatiotemporal differences in exponentially decreasing $T_B$ signatures over the percolation facies.
We have found that by correlating exponentially decreasing $T_B$ signatures with perennial firn
aquifer areas identified via airborne ice-penetrating radar surveys that this algorithm can be
effectively calibrated.

While in this study we converted the exponential rate of $T_B$ decrease to a simple binary mapping
parameter and normalized $T_B$ time series, an improved understanding of the physics controlling
L-band perennial firn aquifer emissions is critical for the development of more sophisticated
retrieval techniques to map other parameters, such as physical temperature (Jezek et al., 2015)
and depth to the upper surface of stored meltwater. Depth is a key control on meltwater-driven
hydrofracture (van der Veen, 2007) and thus, the retrieval of this parameter from space has
important implications for monitoring ice sheet-wide instability and ongoing mass loss.



Perennial firn aquifers represent a radiometrically cold subsurface reservoir, similar to subglacial
lakes, as suggested by Jezek et al., (2015). Field measurements suggest that the volumetric
fraction of meltwater within Greenland's perennial firn aquifers may be as high as ~25% (Koenig
et al., 2014), resulting in high permittivity ($\varepsilon_r \approx 9 + 1i$) which limits the transmission of
electromagnetic radiation (~10%). Upwelling emissions from deeper glacial ice, and from below
the upper surface of stored meltwater, are extinguished by reflection at water-saturated layer
interfaces. While radiometrically cold, the slow refreezing of deeper firn layers saturated with large
volumetric fractions of meltwater represents a significant source of latent heat that is continuously
released throughout the freezing season. Refreezing of seasonal meltwater by the descending
winter cold wave, and the subsequent formation of embedded ice structures within the upper firn
layers, represents a secondary source of latent heat. Perennial firn aquifer areas are physically,
and thus radiometrically, warmer than other percolation facies areas where the single source of
latent heat is via refreezing of seasonal meltwater, and the formation of embedded ice structures.
We hypothesize the key control on the relatively slow exponential rate of $T_B$ decrease in perennial
firn aquifer areas is physical temperature at depth. Emissions from radiometrically warm firn layers
are decreased over time as embedded ice structures slowly refreeze at increased depths below
the ice sheet surface and induce strong volume scattering. Simulating the exponential rate of $T_B$
decrease and the associated changes in $T_B$ magnitudes over perennial firn aquifer areas by
combining electromagnetic forward models that include embedded ice structure parametrizations
(Jezek et al., 2018) with plausible models of depth dependent physical properties can test this
hypothesis and is the focus of ongoing studies. Of particular interest is understanding the
relationship between the exponential rate of $T_B$ decrease, and changes in the depth to the upper
surface of stored meltwater over time.





*Data availability.*

Enhanced-resolution L-band $T_B$ image time series have been produced as part of the NASA Science Utilization of SMAP project and are available at https://doi.org/10.5067/QZ3WJNOUZLFK. IST image time series have been produced as part of the Multilayer Greenland Ice Surface Temperature, Surface Albedo, and Water Vapor from MODIS V001 data set and are available at https://doi.org/10.5067/7THUWT9NMPDK. The NASA MEaSURE's Greenland Ice Mapping Project (GIMP) Land Ice and Ocean Classification Mask, Version 1 is available at https://doi.org/10.5067/B8X58MQBFUPA. The NASA MEaSUREs ALOS-PALSAR winter season radar backscatter ($\sigma^o$) image mosaic is available at https://doi.org/10.5067/6187DQUL3FR5. The coastline data is available from GSHHG - A Global Self-consistent, Hierarchical, High-resolution Geography Database http://marine.gov.scot/data/gshhg-global-self-consistent-hierarchical-high-resolution-geography-database.

*Author contributions.*

JZM initiated the study, performed the analyses, and wrote the manuscript. DGL and MJB generated the enhanced-resolution L-band $T_B$ image time series. JZM and DGL developed the empirical algorithm. CAS provided perspective on the IST data. KCJ and JZM imagined the emissions concept. All authors reviewed and commented on manuscript drafts.

*Competing interests.*

The authors declare that they have no competing interests.

*Acknowledgments.*

JZM would like to thank the Byrd Postdoctoral Fellowship Program. This work was supported by the NASA Instrument Incubator Program (grant #NNX14AE68G) and the NASA Cryospheric Science Program (#80NSSC18K0550) under grants to The Ohio State University, and by the NASA Cryospheric Science Program (#80NSSC18K1055) and the NSF Antarctic Glaciology Program (# PLR-1745116) under grants to the University of Colorado. We would like to thank Dorothy Hall for providing the IST image time series very early in the study, and Clement Miège for providing the MCoRDS- and Accumulation Radar-derived perennial firn aquifer locations.

Development and production of the enhanced-resolution SMAP data used for this research was funded by the NASA ROSES Program Element "Science Utilization of Soil Moisture Active





Passive Mission" under grants to Brigham Young University (#NNX16AN01G) and the University of Colorado at Boulder (#NNX16AN02G).

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
