# Peer review of "Brief communication"

_The Cryosphere, 2020_

## Referee Comment (RC1) · Anonymous Referee #1 · 10 Mar 2020

**Review of *Mapping Greenland's perennial firn aquifers using enhanced resolution L-band brightness temperature image time series***

Summary

In this manuscript, an empirical model is developed to map the extent of firn aquifers over the Greenland ice sheet using L-band brightness temperatures time series. L-band signatures of firn aquifers are shown to exhibit a slow exponential decrease before reaching stable magnitudes late in the freezing season in contrast to non-firn aquifer areas with rapid exponential decrease. Sigmoidal curves are fitted to the L-band time series and the classification of firn aquifers is based on the modelled rate of decrease. The empirical algorithm developed is calibrated using airborne surveys of firn aquifers. Using this new method, the authors found that perennial firn aquifers are found over 66, 000 km$^2$ of the Greenland ice sheet.

General Comments

This is an interesting paper using L-band brightness temperatures from the SMAP satellite to map perennial firn aquifers. The paper is well-written and the empirical method developed here seems to provide a consistent mapping with airborne surveys of aquifers. However, there are a few points that I think would substantially improve the paper:

1) On the calibration of the model, you need to explain why you decided on the interval $\zeta \in$ [-0.04;-0.008] to produce the final map. This is very important as your mapping is dependent on the interval chosen for the rate of decrease.
2) I think that the paper is missing a quantitative comparison to previous studies. Based on the present form of the manuscript, it is hard to evaluate how the new method developed here performs compared to other surveys of firn aquifers. Despite Figure 3 showing a consistent pattern with previous airborne surveys (which is expected as the model developed here is calibrated using these airborne observations), the paper lacks some form of quantitative analysis of the performance of the model.
3) I find that Figures 1b&d and 3b&d lack clarity. The legends of Figure 1 and 3 need some clarifications and the authors need to check the consistency between the colours used in Figure 2 and what is stated in the text (see comments below).

I think that the manuscript mainly requires clarifications of the points above and I therefore recommend the paper to be published after minor revisions. I expect that this paper will be an interesting addition for studies of firn aquifers and will be of interest for the scientific community.

Specific Comments

**L66:** This newly published paper could be of interest here as it uses Sentinel-1 to map firn aquifers from space: Brangers, I., Lievens, H., Miège, C., Demuzere, M., Brucker, L., & De Lannoy, G. J. M. (2020). Sentinel-1 detects firn aquifers in the Greenland ice sheet. *Geophysical Research Letters*, 47, e2019GL085192. https://doi.org/10.1029/2019GL085192

**L90:** It would be helpful if you could state here what the resolution needed to map firn aquifers is and in how many passes this can typically be achieved

**L233:** I would say in steps of 0.004 rather than in time steps of 0.004 which is confusing as you increase the exponential rate and not the time parameter

**L235-237:** How did you choose the calibration interval [-0.04;-0.008]? You need to explain how you came to choose these values as the final binary maps are based on this interval. For instance you could provide a histogram of the exponential decrease rate you obtain for your whole set of time series.

**L237 and L253:** typo [-1;-0.04]

**L242:** What about H-pol time series? You mention at L186 that you analysed the H-pol time series as well but you only fit the V-pol time series?

**L267:** You state that the number of iterations is between 1 and 10 but based on Figure 2, the number of iterations seems to be higher (21 at sites 3 and 4)?

**L273:** Based on Figure 2, the colours corresponding to sites 1-4 are blue, cyan, red and green and not blue, teal, yellow and green

**L278:** In the text, Test Sites 4 and 5 are reported as marked with orange and red lines but in Figure 2, Test Sites 4 and 5 are marked by the green and orange lines. Don't you mean Test Sites 3 and 5 instead (see my comment below)?

**L273-280:** Aren't Test Sites 3 and 5 (red and orange lines) examples of other facies areas and Test Sites 1, 2, 4 and 6 sites with firn aquifer signatures?

**L311:** Please state how many grid cells.

**L287:** It would be interesting to comment on how the firn aquifer extent you derived from L-band data relates to previously reported estimates of firn aquifer extent

**L306-320:** Please be more specific in your comparison to MCoRDS and Accumulation Radar-derived firn aquifers and provide some quantitative metrics.

**Figure 1:** In the caption, I think that what is labelled as Figure b) is actually Figure c). I think that it would be best to change the colour of Test Site 1 for something different than blue as the MCoRDS-derived firn aquifers are also marked in blue (same for Figure 3). Also what is the difference between the yellow lines in b) and the black lines in d)?

**Figure 3:** Idem as in Figure 1, Figure c) is labelled as Figure b here. I think that you need to clarify the legends of Figure b) and d). I guess that MCoRDS-derived firn aquifers are in blue and not in yellow and blue circles? I find it hard to understand what you are showing on b) and d)

---

## Referee Comment (RC2) · Anonymous Referee #2 · 30 Mar 2020

Mapping Greenland's perennial firn aquifers using enhanced-resolution L-band brightness temperature image time series

By J. Miller et al.

**General comments**

This is an exciting study, introducing a new method to map the Greenland firn aquifers from passive microwave L-band satellite observations. The study is very relevant, well-written and well-organized. I'd recommend its publication, but some important aspects first need to be addressed:

1. As also pointed out by Reviewer #1, the study lacks a clear description of the calibration strategy, and also a quantitative validation is lacking. The airborne OIB data set could for instance be split into calibration and validation subsets, in order to perform an independent performance assessment of the aquifer mapping. As it reads, it seems that the authors selected the range in ζ of [-0.04 -0.008] only based on a visual inspection of the resulting aquifer maps, in comparison with maps derived from OIB. But, the range in ζ will have a strong impact on the total aquifer extent and should thus be selected with care. In addition to a better developed calibration strategy, I'd also recommend to perform and show a sensitivity analysis of the total aquifer extent in function of the limits of ζ.
2. I miss a critical discussion by the authors of what they perceive as the strengths and weaknesses or uncertainties of their approach, and where they expect it to work well and less well. The first paragraph on page 12 (L309-L315) goes somewhat into that direction. But this discussion could be improved. For instance, are there brightness temperature signatures of non-aquifer areas that could potentially be confused with those of aquifer areas? Which areas? Or vice-versa, do some aquifer areas lack the characteristic behavior, and why? What could be the impact of seasonal meteorological conditions on the aquifer (or non-aquifer) signatures, etc.

Specific comments

1. L62: Perhaps explicitly mention that there have been flights during 2015-2019 as well? Have you considered including one or more additional years, which could be helpful for testing the robustness of the method?
2. L82-90: Would it be possible to provide an indication of the effective spatial resolution, resulting from this processing? Do you expect that the relatively coarse spatial resolution of passive L-band microwave could have a strong impact on the total aquifer extent? For instance, could the extent possibly be overestimated due to the coarse resolution, given that several pixels may only partially be covering an aquifer in reality (while I realize that also some pixels may not be classified as aquifer due to mixing signals from non-aquifer fractions)?
3. L102: I believe this is the first mentioning of OIB data being available after 2014? (see also specific comment 1).

4.  L105-120: In my opinion, it would be far more interesting for this particular study to delete the entire section discussing active microwave signatures over Greenland from PALSAR (L107-118) and replace that by a section which explains more into detail the signatures in brightness temperature for different facies of the Greenland ice sheet. These passive microwave signatures could later on help supporting a discussion of strengths and uncertainties of the aquifer detection method. Active microwave signatures (the focus of this section in its present form) are nowhere used in the method, validation, or analysis; only as a background in some of the maps.
5.  L162: How have IST data been projected and upscaled onto the EASE-2 grid? By linear averaging?
6.  L165-168: are there any references to support these assumptions?
7.  Section 2.3: The 2016 aquifer picks were based on the MCoRDS instrument. This instrument is less well suited for aquifer delineation than the OIB accumulation radar flown in several other campaign years, and may potentially lead to biases (likely under-detection)? Perhaps this is worth mentioning when comparing your classification with OIB? Also, including some of these other years may improve the robustness of the calibration of your method.
8.  Section 2.4.1: This explanation is very similar to that in a recent study published by Brangers et al. (2020) in GRL, discussing signatures of active microwave (Sentinel-1) for mapping Greenland firn aquifers. Perhaps it is worth mentioning this similarity, to provide additional support for your method.
9.  Section 2.4.1: I would suggest to move the section on page 13 (L338-350) to somewhere around section 2.4.1 within the methods, since this provides the theoretical support for your classification method. Moreover, it is not well placed in the summary and future work section, since it provides new theoretical information (not a summary).
10. L228-239: Which regions typically correspond to $\zeta > -0.008$ and why?
11. L244: Decreased sensitivity: do you mean relative to H-pol?
12. L289-293: Some of these aquifer locations are also revealed in the study of Brangers et al. (2020). Perhaps it'd be interesting to compare some of your results (such as total aquifer area) with that study?
13. The paper often refers to 'perennial' firn aquifers. How can you be sure that the firn aquifers in some places are not completely refrozen late in the frozen season, based on your detection method?

Technical corrections

1.  Please check figure color references, subpanel references etc. throughout the manuscript.
2.  L78: SMAP was launched on January 31.
3.  L115: The range in wavelengths seems too wide for L-band only?
4.  Figure 1c,d: A color scale is lacking. Maybe passive microwave data, or a DEM provide a more suitable background than PALSAR?
5.  Figure 2a: Minus signs before the values of dzeta are missing
6.  Data availability: the last link to the coastline data does not work (when I tried on my laptop).

---

## Author Comment (AC1) · 10 May 2020

Authors Response to:

**Review of Mapping Greenland's perennial firn aquifers using enhanced resolution L-band brightness temperature image time series**

Thank you for your thoughtful comments!

**Specific Comments L66: This newly published paper could be of interest here as it uses Sentinel-1 to map firn aquifers from space: Brangers, I., Lievens, H., Miège, C., Demuzere, M., Brucker, L., & De Lannoy, G. J. M. (2020). Sentinel-1 detects firn aquifers in the Greenland ice sheet. Geophysical Research Letters, 47, e2019GL085192. https://doi.org/10.1029/2019GL085192.**

Added citations. See L60 and L319.

**L90: It would be helpful if you could state here what the resolution needed to map firn aquifers is and in how many passes this can typically be achieved**

Not enough is known about perennial firn aquifers to answer the question of resolution. The airborne observations are far too sparse, and leave many unanswered questions about the behavior of firn aquifers in the boundary regions and between flight lines.

L-band is the only frequency with a penetration depth capable of directly observing the upper surface of meltwater throughout the winter. However the resolution is relatively coarse (as compared to say, a SAR) and the record only starts recently (SMOS, 2009). Given the shallow penetration depth (~several meters) C-band can only observe meltwater for a few months, at most, after surface melting ends and thus an inference is made which leaves uncertainty the ultimate fate of meltwater (perennial?) in the mapping. However, there is a record that dates back to ~1992 (AMI, ERS satellite series). Mapping is really dependent on the availability of satellite data, rather than a specific resolution.

**L233: I would say in steps of 0.004 rather than in time steps of 0.004 which is confusing as you increase the exponential rate and not the time parameter**

Revised in text. See L233-234

*incremented by steps of 0.004*

**L235-237: How did you choose the calibration interval [-0.04;-0.008]? You need to explain how you came to choose these values as the final binary maps are based on this interval.**

Revised in text. See L254-267.

*To distinguish between perennial firn aquifer areas and percolation facies areas where seasonal meltwater is refrozen and stored exclusively as embedded ice, we calibrated the curve fitting algorithm using the MCoRDS-derived perennial firn aquifer locations projected on the EASE-Grid 2.0. For each grid cell we extracted V-pol $T_B$ time series and ζ values, and for each of these calibration parameters we calculated the standard deviation (σ). We set thresholds of ±2σ in an attempt to eliminate peripheral grid cells near the ice edge and near the upper perennial firn aquifer boundary where L-band emissions are influenced by morphological features, such as crevasses and exposed glacial ice, and mix with emissions from rock, land, the ocean, and adjacent percolation facies areas. We set a*

*minimum brightness temperature threshold of $T_{min}$=200, and a maximum brightness temperature threshold of $T_{max}$=240, and an exponential rate of normalized brightness temperature decrease threshold of $\zeta \in [-0.04, -0.008]$. If the calibration parameters are within the threshold intervals, the grid cell is converted to a simple binary parameter to map extent. $T_B$ time series iteratively fit to the sigmoid function converge quickly (i.e., algorithm iterations $I \in [5, 19]$), and observations are a good fit (i.e., chi squared error statistic is $\chi 2 \in [0, 0.06]$), indicating our algorithm provides a plausible satellite-derived map of the extent of Greenland's perennial firn aquifers.*

**L237 and L253: typo [-1;-0.04]**

Corrected in text and figures.

***L242: What about H-pol time series? You mention at L186 that you analysed the H-pol time series as well but you only fit the V-pol time series?***

Clarified in text. See L242-247.

*The curve fitting algorithm proceeds by smoothing V-pol $T_B$ time series that have been partitioned by TIR $T_B$-derived surface freeze-up and melt onset dates, and then iteratively applying the sigmoid fit. This results in a reduced chi-squared error statistic when fitting V-pol $T_B$ time series to the sigmoid function. The V-pol channel exhibits decreased sensitivity to changes in the volumetric fraction of meltwater as compared to the H-pol channel. We attribute these differences to reflection coefficient differences between channels. We note, however, that both channels provide reasonable results.*

**L267: You state that the number of iterations is between 1 and 10 but based on Figure 2, the number of iterations seems to be higher (21 at sites 3 and 4)?**

Thank you for pointing this out. Figure 2 revised.

**L273: Based on Figure 2, the colours corresponding to sites 1-4 are blue, cyan, red and green and not blue, teal, yellow and green L278: In the text, Test Sites 4 and 5 are reported as marked with orange and red lines but in Figure 2, Test Sites 4 and 5 are marked by the green and orange lines. Don't you mean Test Sites 3 and 5 instead (see my comment below)?**

Thank you for pointing this out. Figure 2 revised.

***L273-280: Aren't Test Sites 3 and 5 (red and orange lines) examples of other facies areas and Test Sites 1, 2, 4 and 6 sites with firn aquifer signatures?***

Revised in text.

***L311: Please state how many grid cells.***

Text removed.

***L287: It would be interesting to comment on how the firn aquifer extent you derived from L-band data relates to previously reported estimates of firn aquifer extent***

Revised text. See L312-319.

*The L-band $T_B$-derived perennial firn aquifer extent is generally consistent with previous C-band (5.3 GHz) satellite radar scatterometer-derived extents mapped using the Advanced SCATterometer (ASCAT) on the European Organization for the Exploitation of Meteorological Satellites (EUMETSAT) Meteorological Operational A (MetOp-A) satellite (2009-2016, ~52,000 km–153,000 km, Miller, 2019), and the Active Microwave Instrument in scatterometer mode (ESCAT) on ESA's European Remote Sensing (ERS) satellite series (1992-2001, ~37,000 km-64,000 km, Miller, 2019) as well as with the C-band (5.4 GHz) synthetic aperture radar-derived extent mapped using the synthetic aperture radar on ESA's Sentinel-1 satellite (2014-2019, 54,000, Brangers et al., 2020).*

***L306-320: Please be more specific in your comparison to MCoRDS and Accumulation Radar-derived firn aquifers and provide some quantitative metrics.***

This is not a reasonable comparison given the small size of the MCoRDS locations and the footprint of the scatterometer.

See line 91-100

*Given converging orbital passes in the polar regions, the SMAP satellite passes over Greenland several times each day, and provides nearly complete coverage during two distinct local time-of-day intervals. The rSIR algorithm combines orbital passes that occur between 8 a.m. and 4 p.m. (+/-2 hours) local time-of-day to reconstruct twice-daily (morning and evening orbital pass interval, respectively) $T_B$ images. $T_B$ image data are projected on the Equal-Area Scalable Earth Grid (EASE-Grid 2.0) (Brodzik et al., 2012) at a 3.125 km posting resolution or grid cell spacing. The effective resolution for each grid cell is dependent on the number of observations used in the rSIR reconstruction and is coarser than the grid cell spacing. While the effective resolution of conventionally processed $T_B$ images posted on a 25 km grid is ~30 km, the effective resolution of enhanced resolution* **$T_B$ images posted on a 3.125 km grid is ~18 km**, *an improvement of ~60%.*

See L170-176

*We projected the MCoRDS-derived perennial firn aquifer locations on the EASE-Grid 2.0 at a grid cell spacing of 3.125 km. Each grid cell has an extent of ~10 km. The total number of grid cells with at least one location is 780 corresponding to an extent of ~7617 km²; however, less than ~5% of this extent is an actual detection. The maximum number of detections in a grid cell is 50, corresponding to an extent of 0.25 km² or ~8% of a grid cell. These three locations are along crossing flight lines near Test Site 1 (Fig. 1). The remaining detections are along linear flight lines. The mean number of detections in a grid cell is 18, corresponding to 0.09 km² or ~1% of a grid cell.*

See L269-276

*We note, however, that the lack of a distinct L-band $T_B$ signature that delineates the boundary between perennial firn aquifer areas and adjacent percolation facies areas and the limited number of MCoRDS-derived perennial firn aquifer locations results in significant uncertainty in the mapped extent. If V-pol $T_B$ time series are not quite within the calibration intervals, it does not necessarily indicate that a perennial firn aquifer is not present over at least a percentage of a grid cell. A sensitivity analysis suggests that even small changes in the calibration intervals (i.e., a few K for $T_{min}$ and $T_{max}$ values, and a few hundredths of a percentage point for ζ values) can result in extent changes of hundreds of kilometers. Thus, the mapped extent should simply be considered a rough estimate.*

*Figure 1: In the caption, I think that what is labelled as Figure b) is actually Figure c). I think that it would be best to change the colour of Test Site 1 for something different than blue as the MCoRDS derived firn aquifers are also marked in blue (same for Figure 3). Also what is the difference between the yellow lines in b) and the black lines in d)? Figure 3: Idem as in Figure 1, Figure c) is labelled as Figure b here. I think that you need to clarify the legends of Figure b) and d). I guess that MCoRDS-derived firn aquifers are in blue and not in yellow and blue circles? I find it hard to understand what you are showing on b) and d)*

Figure 1 and 3 revised.

---

## Author Comment (AC2) · 10 May 2020

**Author response to Review 2:**

**Mapping Greenland's perennial firn aquifers using enhanced-resolution L-band brightness temperature image time series**

**1. As also pointed out by Reviewer #1, the study lacks a clear description of the calibration strategy, and also a quantitative validation is lacking. The airborne OIB data set could for instance be split into calibration and validation subsets, in order to perform an independent performance assessment of the aquifer mapping. As it reads, it seems that the authors selected the range in ζ of [-0.04 -0.008] only based on a visual inspection of the resulting aquifer maps, in comparison with maps derived from OIB. But, the range in ζ will have a strong impact on the total aquifer extent and should thus be selected with care. In addition to a better developed calibration strategy, I'd also recommend to perform and show a sensitivity analysis of the total aquifer extent in function of the limits of ζ.**

Revised in the text. See 2 below. There is simply no way to quantitatively compare the microwave radiometer footprint with an effective resolution of 18 km and the sparse airborne footprint which cover between 1% and 8% of a very limited number a grid cells. Although C-band comparison may provide some insight, given the shallow penetration depth (~several meters) C-band can only observe meltwater for a few months, at most, after surface melting ends and thus an inference is made which leaves uncertainty the ultimate fate of meltwater (perennial?) in the mapping. That is a possibility, but beyond the scope of this very short paper and saved for future work.

**2. I miss a critical discussion by the authors of what they perceive as the strengths and weaknesses or uncertainties of their approach, and where they expect it to work well and less well. The first paragraph on page 12 (L309-L315) goes somewhat into that direction. But this discussion could be improved. For instance, are there brightness temperature signatures of non-aquifer areas that could potentially be confused with those of aquifer areas? Which areas? Or vice-versa, do some aquifer areas lack the characteristic behavior, and why? What could be the impact of seasonal meteorological conditions on the aquifer (or non-aquifer) signatures, etc.**

Revised in the text. See 10, 11, and 13 below. This is a very short paper, and there simply was not room to detail the range of signatures observed over the Greenland Ice Sheet – which is a complicated discussion. This is a very good path forward, but saved for future work (and a longer paper!). ☺

**Specific comments**

**1. L62: Perhaps explicitly mention that there have been flights during 2015-2019 as well? Have you considered including one or more additional years, which could be helpful for testing the robustness of the method?**

**7. Section 2.3: The 2016 aquifer picks were based on the MCoRDS instrument. This instrument is less well suited for aquifer delineation than the OIB accumulation radar flown in several other campaign years, and may potentially lead to biases (likely underdetection)? Perhaps this is worth mentioning when comparing your classification with OIB? Also, including some of these other years may improve the robustness of the calibration of your method.**

There is only one other airborne (MCoRDS) data set coincident with SMAP (2017) – which would have made two years. Because this was a short paper, we decided to include only one year. The calibration thresholds change between years, which would have needed additional discussion that simply would not fit in this short paper. It is saved for future work. ;)

It's very likely that MCoRDS is less robust, But, the sparsity of the airborne data is the key uncertainty (described in the next section)

**3. L102: I believe this is the first mentioning of OIB data being available after 2014? (see also specific comment 1).**

Text Revised. See L49-54.

*The existence and approximate extent of Greenland's perennial firn aquifers has been demonstrated using shallow firn cores and ground penetrating radar surveys collected at several sites in southeastern Greenland during recent field expeditions, (Forster et al., 2014; Koenig et al., 2014; Miller et al., 2017) as well as locations detected using ice-penetrating radar surveys collected by the CreSIS Accumulation Radar flown by NASA's OIB campaign (Miège et al., 2016). Additional locations have more recently been detected using ice penetrating radar surveys collected by the MCoRDS instrument.*

***2. L82-90: Would it be possible to provide an indication of the effective spatial resolution, resulting from this processing? Do you expect that the relatively coarse spatial resolution of passive L-band microwave could have a strong impact on the total aquifer extent? For instance, could the extent possibly be overestimated due to the coarse resolution, given that several pixels may only partially be covering an aquifer in reality (while I realize that also some pixels may not be classified as aquifer due to mixing signals from non-aquifer fractions)?***

Text revised.

See L94-100

$T_B$ *image data are projected on the Equal-Area Scalable Earth Grid (EASE-Grid 2.0) (Brodzik et al., 2012) at a 3.125 km posting resolution or grid cell spacing. The effective resolution for each grid cell is dependent on the number of observations used in the rSIR reconstruction and is coarser than the grid cell spacing. While the effective resolution of conventionally processed $T_B$ images posted on a 25 km grid is ~30 km, the effective resolution of enhanced resolution $T_B$ images posted on a 3.125 km grid is ~18 km, an improvement of ~60%.*

The resolution (and the sparsity of airborne locations) absolutely has an influence on the extent!

See lines L170-176

*We projected the MCoRDS-derived perennial firn aquifer locations on the EASE-Grid 2.0 at a grid cell spacing of 3.125 km. Each grid cell has an extent of ~10 km. The total number of grid cells with at least one location is 780 corresponding to an extent of ~7617 km²; however, less than ~5% of this extent is an actual detection. The maximum number of detections in a grid cell is 50, corresponding to an extent of 0.25 km² or ~8% of a grid cell. These three locations are along crossing flight lines near Test Site 1 (Fig. 1). The remaining detections are along linear flight lines. The mean number of detections in a grid cell is 18, corresponding to 0.09 km² or ~1% of a grid cell.*

See L269-276

*We note, however, that the lack of a distinct L-band $T_B$ signature that delineates the boundary between perennial firn aquifer areas and adjacent percolation facies areas and the limited number of MCoRDS-derived perennial firn aquifer locations results in significant uncertainty in the mapped extent. If V-pol $T_B$ time series are not quite within the calibration intervals, it does not necessarily indicate that a perennial firn aquifer is not present over at least a percentage of a grid cell. A sensitivity analysis suggests that even small changes in the calibration intervals (i.e., a few K for $T_{min}$ and $T_{max}$ values, and a few hundredths of a*

*percentage point for ζ values) can result in extent changes of hundreds of kilometers. Thus, the mapped extent should simply be considered a rough estimate.*

**4. L105-120: In my opinion, it would be far more interesting for this particular study to delete the entire section discussing active microwave signatures over Greenland from PALSAR (L107-118) and replace that by a section which explains more into detail the signatures in brightness temperature for different facies of the Greenland ice sheet. These passive microwave signatures could later on help supporting a discussion of strengths and uncertainties of the aquifer detection method. Active microwave signatures (the focus of this section in its present form) are nowhere used in the method, validation, or analysis; only as a background in some of the maps.**

PALSAR section removed, and a short section (that fits within the text requirements) on L-band signatures is added.

*See L180-193*

*We analyzed V- and H-pol $T_B$ time series over and around the MCoRDS-derived perennial firn aquifer locations projected on the EASE-Grid 2.0. These time series were overlaid with TIR $T_B$-derived surface freeze-up and melt onset dates to partition the freezing season. Throughout Greenland's percolation facies, $T_B$ magnitudes over perennial firn aquifer areas are radiometrically warm, ranging from ~200 K to 230 K (V-pol channel) and ~160 K to 200 K (H-pol channel). L-band $T_B$ signatures exhibit relatively slow (i.e., time scales of ~months) exponential decreases that approach or achieve relatively stable $T_B$ magnitudes late in the freezing season. Exponential decreases are the slowest in the physically warmer southern regions of the Greenland Ice Sheet, and increase moving toward the colder northern regions. In contrast, $T_B$ magnitudes over other percolation facies areas where seasonal meltwater is refrozen and stored exclusively as embedded ice are radiometrically colder, ranging from ~130 K to 200 K (V-pol channel) and ~ 100 to 160 K (H-pol channel). L-band $T_B$ signatures exhibit relatively rapid (i.e., time scales of ~weeks to days) exponential decreases, subsequently achieve relatively stable $T_B$ magnitudes early in the freezing season, and remain relatively stable until melt onset the following year. Exponentially decreasing signatures transition smoothly between these two areas – there is no distinct $T_B$ signature that delineates a boundary.*

**5. L162: How have IST data been projected and upscaled onto the EASE-2 grid? By linear averaging?**

Text revised.

See L146 and 147.

*We projected the IST image data onto the EASE-Grid 2.0 at a 3.125 km grid cell spacing using rigorous orthorectification and cubic convolution,*

**6. L165-168: are there any references to support these assumptions?**

Reference added.

See L149-151

*We set a threshold of IST ≥ -1°C for surface meltwater detection (Nghiem et al., 2012), consistent with the ±1°C accuracy of the IST image data (i.e., surface meltwater is inferred when IST is as low as -1°C). This threshold represents a penetration depth from ~2 μm beneath the snow.*

**8. Section 2.4.1: This explanation is very similar to that in a recent study published by Brangers et al. (2020) in GRL, discussing signatures of active microwave (Sentinel-1) for mapping Greenland firn aquifers. Perhaps it is worth mentioning this similarity, to provide additional support for your method.**

**12. L289-293: Some of these aquifer locations are also revealed in the study of Brangers et al. (2020). Perhaps it'd be interesting to compare some of your results (such as total aquifer area) with that study?**

The authors have actually done extensive C-band radar (scatterometry) work – and first identified that signature soon after the aquifer was discovered (lead author was at the University of Utah). Both of these works are cited – but not described in detail given the lack of space. This is also the focus of future work.

See L66-74

*Initial studies have shown that C-band radar backscatter collected by satellite radar scatterometers (Miller et al., 2013), and, more recently, by satellite synthetic aperture radar (Brangers et al., 2020), are sensitive to subsurface meltwater storage in the upper snow and firn layers. However, the C-band penetration depth in the frozen snow and firn layers of Greenland's percolation facies is on the order of several meters, and the mean depth of the upper surface of meltwater in Greenland's perennial firn aquifers just prior to melt onset is estimated to be ~22 m (Mìège et al., 2016).*

and L312-319

*The L-band T_B-derived perennial firn aquifer extent is generally consistent with previous C-band (5.3 GHz) satellite radar scatterometer-derived extents mapped using the Advanced SCATterometer (ASCAT) on the European Organization for the Exploitation of Meteorological Satellites (EUMETSAT) Meteorological Operational A (MetOp-A) satellite (2009-2016, ~52,000 km–153,000 km, Miller, 2019), and the Active Microwave Instrument in scatterometer mode (ESCAT) on ESA's European Remote Sensing (ERS) satellite series (1992-2001, ~37,000 km-64,000 km, Miller, 2019) as well as with the C-band (5.4 GHz) synthetic aperture radar-derived extent mapped using the synthetic aperture radar on ESA's Sentinel-1 satellite (2014-2019, 54,000, Brangers et al., 2020).*

**9. Section 2.4.1: I would suggest to move the section on page 13 (L338-350) to somewhere around section 2.4.1 within the methods, since this provides the theoretical support for your classification method. Moreover, it is not well placed in the summary and future work section, since it provides new theoretical information (not a summary).**

Agreed. The authors changed the section to **4. Discussion and Future Work** – rather than a summary. As this is mostly an empirical study, the theoretical description was included at the end as a path to our next paper which explores both annual and perennial firn aquifer signatures using field data and electromagnetic modeling.

**10. L228-239: Which regions typically correspond to ζ > -0.008 and why?**

The authors originally assumed this lower threshold corresponded to shallow firn aquifers in the peripheral areas. However, these signatures are typically radiometrically warm and a shallow heavily water saturated perennial firn aquifer would be radiometrically cold. A better explanation might be that these signatures are mixed emission firn aquifer-ice signatures, where the ice is the peripherally warm emitter. Another explanation is that there is a heavily saturated layer with an overlying layer with lower volumetric fractions of meltwater throughout the winter (the warm emitter). It's difficult to know – and there are many signatures with similarly complicated explanations, which is why a detailed explanation was left out and saved for future work.

**11. L244: Decreased sensitivity: do you mean relative to H-pol?**

Clarified in text.

See L244-247

*The V-pol channel exhibits decreased sensitivity to changes in the volumetric fraction of meltwater as compared to the H-pol channel. We attribute these differences to reflection coefficient differences between channels. We note, however, that both channels provide reasonable results.*

**13. The paper often refers to 'perennial' firn aquifers. How can you be sure that the firn aquifers in some places are not completely refrozen late in the frozen season, based on your detection method?**

I honestly don't believe there is any way to be sure. If the signature continually decreased over the entire freezing season, then that might be considered a 'for sure' perennial firn aquifer case. But that only happens sometimes, and would need a more sophisticated time series analysis to identify exact dates. However, a perennial firn aquifer can also reach a stable depth at a given point in the freezing season. In this case, the signature becomes stable. However, an aquifer that completely refreezes also becomes stable. This may be able to be somewhat sorted out with brightness temperature values, however, these vary in both space and time. Although the authors have done extensive analysis on this topic, we still don't have a good grasp on the behavior of these signatures. They are extremely complicated. And left to future work.

**Technical corrections**

**1. Please check figure color references, subpanel references etc. throughout the manuscript.**

Checked and corrected. Thank You.

**2. L78: SMAP was launched on January 31.**

Corrected.

**3. L115: The range in wavelengths seems too wide for L-band only?**

This text was actually referring to the size of the pipes and lenses, but I can see how it was confusing. This text was moved to a different section and revised.

See L351-356

*We hypothesize the key control on the relatively slow exponential rate of $T_B$ decrease in perennial firn aquifer areas is physical temperature at depth. L-band emissions from radiometrically warm firn layers are decreased over time as embedded ice structures slowly refreeze at increased depths below the ice sheet surface. Refreezing of seasonal meltwater results in the formation of an intricate network of embedded ice structures (i.e., ice pipes, lenses, and layers) that are large (~10-100 cm long, ~10-20 cm wide, Jezek et al., 1994) relative to the L-band wavelength (21 cm) and induce strong volume scattering (Rignot, 1995).*

**4. Figure 1c,d: A color scale is lacking. Maybe passive microwave data, or a DEM provide a more suitable background than PALSAR?**

Figure 1 was revised to focus a little more on the resolution enhancement. We did replace the PALSAR image with the L-band passive microwave image in Figure 3.

**5. Figure 2a: Minus signs before the values of dzeta are missing**

Corrected.

**6. Data availability: the last link to the coastline data does not work (when I tried on my laptop).**

Corrected. This link should work now.

Thank you for your insightful comments. ☺

---

## Editor Decision (ED1)

Final Editors comments:

Dear authors,

thankyou for your hard work and revisions which have improved what was already a very nice study.

I am happy to accept it for publication with a very minor revision based on the comments I have screenshot aand inserted below. As you noted in point 9 this is an empirical study and the idea is not to develop the theory so much, however in your explanations for point 10 and point 13 I think you make some very interesting and valid points that would be of interest to the general reader, even if they are hard to back up with concrete evidence and for which there would in any case be little space in this paper. I would therefore like to see these elaborated if at all possible in the discussion section. I do not think it will mean adding more than 2-3 sentences as these are speculative points and possibilities for future investigation.

**9. Section 2.4.1: I would suggest to move the section on page 13 (L338-350) to somewhere around section 2.4.1 within the methods, since this provides the theoretical support for your classification method. Moreover, it is not well placed in the summary and future work section, since it provides new theoretical information (not a summary).**

Agreed. The authors changed the section to **4. Discussion and Future Work** – rather than a summary. As this is mostly an empirical study, the theoretical description was included at the end as a path to our next paper which explores both annual and perennial firn aquifer signatures using field data and electromagnetic modeling.

**10. L228-239: Which regions typically correspond to $\zeta$ > -0.008 and why?**

The authors originally assumed this lower threshold corresponded to shallow firn aquifers in the peripheral areas. However, these signatures are typically radiometrically warm and a shallow heavily water saturated perennial firn aquifer would be radiometrically cold. A better explanation might be that these signatures are mixed emission firn aquifer-ice signatures, where the ice is the peripherally warm emitter. Another explanation is that there is a heavily saturated layer with an overlying layer with lower volumetric fractions of meltwater throughout the winter (the warm emitter). It's difficult to know – and there are many signatures with similarly complicated explanations, which is why a detailed explanation was left out and saved for future work.

**13. The paper often refers to 'perennial' firn aquifers. How can you be sure that the firn aquifers in some places are not completely refrozen late in the frozen season, based on your detection method?**

I honestly don't believe there is any way to be sure. If the signature continually decreased over the entire freezing season, then that might be considered a 'for sure' perennial firn aquifer case. But that only happens sometimes, and would need a more sophisticated time series analysis to identify exact dates. However, a perennial firn aquifer can also reach a stable depth at a given point in the freezing season. In this case, the signature becomes stable. However, an aquifer that completely refreezes also becomes stable. This may be able to be somewhat sorted out with brightness temperature values, however, these vary in both space and time. Although the authors have done extensive analysis on this topic, we still don't have a good grasp on the behavior of these signatures. They are extremely complicated. And left to future work.